# Ten-year trends in clinical characteristics and outcome of children hospitalized with severe wasting or nutritional edema in Malawi (2011–2021): *Declining admissions but worsened clinical profiles*

Mphatso Nancy Chisala[1], Celine Bourdon[2,3]*, Emmanuel Chimwezi[2], Allison I. Daniel[3,4,7], Chikondi Makwinja[2], Dominic Wang[5], Linnea Weise[6], Isabel Potani[2,4,7], Emmie Mbale[8], Robert J. H. Bandsma[2,4,7,9], Wieger P. Voskuijl[2,6,8,10]

1 Department of Population, Policy and Practice, Institute of Child Health, University College London, London, United Kingdom, 2 The Childhood Acute Illness and Nutrition (CHAIN) Network, Nairobi, Kenya, 3 Translational Medicine Program, Hospital for Sick Children, Toronto, Canada, 4 Department of Nutritional Sciences, Temerty Faculty of Medicine, University of Toronto, Toronto, Canada, 5 Schulich School of Medicine and Dentistry, Western University, London, ON, Canada, 6 Amsterdam UMC location University of Amsterdam, Amsterdam Centre for Global Child Health & Emma Children's Hospital, Amsterdam University Medical Centres, Amsterdam, The Netherlands, 7 Center for Global Child Health, Hospital for Sick Children, Toronto, Canada, 8 Department of Pediatrics and Child Health, Kamuzu University of Health Sciences, Blantyre, Malawi, 9 Department of Biomedical Sciences, Kamuzu University of Health Sciences, Blantyre, Malawi, 10 Amsterdam UMC location University of Amsterdam, Amsterdam Institute for Global Health and Development, Amsterdam University Medical Centres, Amsterdam, The Netherlands

☯ These authors contributed equally to this work.
* celine.bourdon@sickkids.ca

## Abstract

### Background

Severe acute malnutrition (SAM) constitutes a substantial burden in African hospitals. Despite adhering to international guidelines, high inpatient mortality rates persist and the underlying contributing factors remain poorly understood.

### Objective

We evaluated the 10-year trend (2011–2021) in clinical factors and outcomes among children with severe wasting and/or nutritional edema at Malawi's largest nutritional rehabilitation unit (NRU).

### Methods

This retrospective study analyzed trends in presentation and outcomes using generalized additive models. The association between clinical characteristics and mortality or readmission was examined and key features were also related to time to either mortality or discharge.

**Data Availability Statement:** All relevant data are within the manuscript and its Supporting Information files.

**Funding:** The author(s) received no specific funding for this work.

**Competing interests:** The authors have declared that no competing interests exist.

## Results

1497 children (53%, females) were included. Median age at admission (23 months, IQR 14, 34) or anthropometry (i.e., weight-for-age, height-for-age and weight-for-height) did not change over the 10-years. But the prevalence of edema decreased by 40% whereas dehydration, difficulty breathing, and pallor became more common. Yearly HIV testing increased but positive-detection remained around 11%. Reporting of complete vaccination dropped by 49%, and no reduction in 'watch' antibiotic usage was detected. Overall admissions declined but mortality remained around 23% [95%CI; 21, 25], and deaths occurred earlier (5.6 days [95%CI; 4.6, 6.9] in 2011 vs. 3.5 days [95%CI; 2.5, 4.7] in 2021; $p$<0.001). Duration of hospitalization was shortened and readmissions surged from 4.9% [95%CI; 3.3, 7.4] in 2011 to 25% [95%CI; 18, 33] in 2021 ($p$<0.001). Age, wasting, having both dehydration and diarrhea, or having vomiting, cough, or difficulty breathing were associated with mortality but these associations did not show any interaction over time.

## Conclusion

Over 10 years, mortality risk remained high among hospitalized children with SAM and coincided with worsened clinical presentation at admission and increased readmission. Longitudinal data from major NRUs can identify shifts in clinical profiles or outcomes, and this information can be leveraged to promote earlier care-seeking, improved risk stratification, and implementation of more patient-centered treatments.

## Introduction

Malnutrition stands as a pivotal risk factor, contributing to 45% of all global under-5 child deaths [1]. Severe acute malnutrition (SAM) includes severe wasting (marasmus), nutritional edema (kwashiorkor) or a mixed phenotype (marasmic-kwashiorkor) [2]. According to World Health Organization (WHO) guidance, children with SAM should be hospitalized if they exhibit one or more Integrated Management of Childhood Illness (IMCI)-danger signs, loss of appetite, and/or severe nutritional edema [3]. Hospitalized patients receive essential care including rehydration fluids, broad-spectrum antibiotics, and protocolized feedings. Despite adherence, critically ill children with severe wasting or nutritional edema are at high risk of mortality which ranges from 10–25% in African hospitals [4–8] and these children face an almost six-fold increase in mortality compared to non-malnourished children [4, 9, 10].

Danger signs are indicators of disease severity and are used for "emergency triage assessment and treatment" for all hospitalized children in low-resource settings, regardless of nutritional status [5, 11, 12]. Structured daily monitoring of danger signs improved mortality prediction for children with SAM hospitalized in three African nutritional rehabilitation units (NRUs) [11]. Recent findings further emphasize their associations with increased mortality risk not only during admission but also in the post-discharge period, highlighting the continued vulnerability of these children [4, 13–15].

Several studies [4, 11, 13, 14, 16–18] and a large meta-analysis [5] have identified clinical factors associated with mortality in hospitalized children with SAM across various low- and middle-income countries (LMICs). Important determinants include edema, low anthropometry, baseline infections (e.g., HIV, diarrhea, pneumonia), and shock [5]. Data suggest reduced yearly mortality of patients with SAM admitted to hospitals [18–20], especially at sites that

implemented guidelines [7], but a large referral facility in South Africa reported persistently high mortality (12%) from 2009 to 2018 [21]. Limited work has aimed to understand changing patterns of readmissions and how clinical presentation may shift over time. The underlying vulnerabilities of children with SAM remain poorly understood and are likely heterogeneous [22] and factors related to mortality may also differ over time with changing clinical presentation and/or co-morbidities and/or access to clinical care [7, 19, 21, 22].

The objective of this study was: 1) to assess the ten-year trends (2011–2021) in admissions, readmissions, and clinical characteristics; 2) to investigate the factors associated with mortality among patients with SAM admitted to the largest NRU in Malawi; and 3) to evaluate if clinical features associated with mortality differ over the 10 year span.

## Material and methods

For this retrospective study, we reviewed the archived clinical charts of children admitted between January 2011 and December 2021 at the Moyo Nutritional Rehabilitation and Research Unit (Moyo NRU) at Queen Elizabeth Central Hospital (QECH), in Malawi. QECH is a national referral centre in Blantyre, with a catchment area of 5 million people in the Southern region of Malawi. Children with SAM were admitted to the Moyo NRU based on contemporary guidelines from the WHO (1999 [23], updated in 2013 [3]) or national Malawian guidelines (implemented in 2016 [24]). Overall, children with SAM were hospitalized if presenting with: i) one or more IMCI-danger signs, ii) an acute medical problem, iii) severe nutritional edema (+++), or iv) with poor appetite.

### Chart selection and data collection

All children with medical charts archived within the 10-year timeframe (2011–2021) were eligible (N = 4,494). Medical files were retrieved for data entry from 8th of June, 2020, to 20th of May, 2022. We followed a systematic sampling technique to select the paper-based records [25]. From files organized by date, a random start point (selected between 1 and 3) was indicated by the data manager. From that point, every 3rd file was flagged for inclusion which were entered by 4 research staff into a customized database built with COMMCARE (https://www.dimagi.com/commcare/features/), an open-access digital data entry platform. Consulted medical files contained patient names, but these were not captured in the database. Date of birth was recorded but deleted after age was calculated. De-identified information included: demographics (i.e., age, sex), breastfeeding and anthropometry (i.e., height, weight, and mid-upper arm circumference (MUAC)). For this, we calculated weight-for-height (or length) z-score (WHZ), weight-for-age z-scores (WAZ), and height (or length)-for-age z-scores (HAZ) using Anthro (ages < 5 years) and AnthroPlus (5–19 years) packages based on 2006 and 2007 WHO growth standards [26]. Severe wasting and/or nutritional edema was defined as per WHO guidelines: Severe wasting, with WHZ < -3 or, if 6–59 months of age, a MUAC < 115 mm; nutritional edema, having bilateral pitting edema only in feet/ankles (Mild, +); in feet and lower limbs, hands or lower arms (Moderate, ++); or generalized edema to feet, legs, hands, arms and face (Severe, +++) [3, 23]. Routinely collected clinical variables were also analysed: fever, vomiting, diarrhea (defined as passing 3 or more loose or liquid stools per day [27]), dehydration (scored as absent, mild, moderate, or severe), convulsion, difficulty breathing, cough, neurological signs, jaundice, pallor, and rash. HIV status was classified as: 'reactive' (i.e., reported HIV positive or with positive rapid diagnosis test and age being > 18 months), 'exposed' (i.e., reactive but < 18 months of age), 'not reactive', or 'untested'. Vaccination status was recorded according to the Malawian extended immunization program as 'complete' -for children with up-to-date vaccinations; 'incomplete' for those who missed/were behind and

'unreported'. We also assessed how antibiotic use changed based on updated WHO AWaRe classification which groups antibiotics into 'Access' (broad-spectrum with a lower risk of resistance, e.g., benzathine penicillin, amoxicillin and chloramphenicol) and 'Watch' (broad-spectrum with higher risk of selection for bacterial resistance, e.g., ceftriaxone, ciprofloxacin and meropenem) [28]. Neurological signs or neuro-disability such as cerebral palsy, and outcomes such as mortality and readmission were also recorded. The collected data pertained to the index admission of the child, while readmission was noted as a dichotomized yes/no variable.

## Statistical analysis

Data analysis was conducted using Stata 17 (Stata Press, StataCorp, 2019) and R (version 4.2.1). Implausible anthropometric values of height ($<$30 cm and age $>$ 12 months, n = 7), weight ($>$ 40 kg, age $<$ 24 months, n = 5) were filtered, and z-scores of $>$4 or $<$-10 were also removed (WHZ, n = 4; HAZ, n = 8; WAZ, n = 1). Descriptive data are presented either as medians and IQR or counts and percentages, split by survival status. Report of breastfeeding was evaluated in children younger than 12 months of age. Changes in the clinical presentation of children admitted to the Moyo NRU over the 10-year period were evaluated using generalized additive models fit with restricted maximum likelihood. Both linear and non-linear trends were tested. The significance of non-linearity was evaluated based on the effective degrees of freedom (EDF) estimated from generalized additive models where an EDF of approximately 1 suggests a linear relationship; an EDF $>$ 1 but $\leq$ 2 suggests a weakly non-linear relationship and an EDF $>$ 2 indicates a highly non-linear relationship that likely has a meaningful inflection point [29]. While complete case analysis and multiple imputation models were considered, these approaches were not chosen. Complete case analysis would have led to significant data loss through listwise deletion and to potential bias due to patterns of missingness not being at random. Imputation of variables with missingness higher than 10–15% and with possible non-random missingness patterns are not considered good candidates for multiple imputation techniques. Thus, we chose a more conservative approach and conducted an "as is" analysed and aimed to assessed reporting biases by evaluating trends in missing values. Model-derived percentages and 95% confidence intervals (95%CI) were calculated at specific years using the emmeans R-package. Clinical characteristics were associated with mortality or readmission using logistic regression models, and their interaction with time was also evaluated. Percentage and 95%CI of mortality or readmission were calculated for each patient group (e.g., WHZ $\geq$ -3.5 or $<$ -3.5; Age $\geq$ 5 years or $\geq$ 2 & $<$ 5 years or $<$ 2 years, edema yes/no; dehydration yes/no, diarrhea yes/no, vomit yes/no, cough yes/no, difficult breathing yes/no, HIV yes/no) and were derived from unadjusted models using the emmeans R-package. Odds ratio (OR) between groups were calculated from models adjusted for age, sex, and HIV status (Level-I adjustment); and additional models were further adjusted for WHZ (Level-II adjustment). We also explored relevant interactions between age and WHZ, and between dehydration and diarrhea or vomiting. Odds ratios (OR) with 95%CI comparing risks of mortality or readmission were derived with the emmeans package. Significant variables ($p<0.05$) were also used to fit competitive risk models, which evaluated differences in time to either mortality or discharge using the cmprsk R-package. This package generates incidence functions that indicate differences in the cumulative probability of either being discharged or dying across hospital stay.

## Ethical approval

Ethical approval for this retrospective analysis was obtained from the National Health Sciences Research Committee in Lilongwe, Malawi (#20/01/2459).

## Results

### 10-year trend indicates evolving clinical profiles of children hospitalized with SAM

From 2011 to 2021, Moyo NRU had 4494 admissions, of which every 3rd file, n = 1498 (33%), were systematically selected for inclusion. One duplicated file was removed leaving a total of 1,497 patient records included in this retrospective analysis (see study flow chart in S1 Fig). The overall clinical characteristics of children at admission are summarized in Table 1 split by survival status while Fig 1 depicts the 10-year trends in clinical presentation at admission. Results are detailed in S1 and S2 Tables. Children had a median age of 23 months (IQR 14, 34), and 776 (53%) were female but there were no discernible changes in age or sex of admitted children over the decade (Fig 1A and S1 Table). Breastfeeding was reported in 106/148 (72%) children under 12 months of age and increased over time (Fig 1B and S1 Table). Overall, 60% [95% CI, 57–62] (i.e., 854/1,497) of children were admitted with nutritional edema but prevalence decreased by approximately 40% over the 10 years (79% [95% CI, 72–85], 19 out of 26, in 2011 to 40% [95% CI, 30–51], 20 out of 49 in 2021) (Fig 1C and S2 Table). Results split by severity of edema are presented in S2 Table and S2 Fig. Anthropometric measures indicated either minimal or non-clinically relevant increases in children with wasting or with nutritional edema over the ten-year span, as illustrated in S3 Fig. However, documentation of anthropometry was relatively poor with missing data in approximately 40% of records. Several IMCI danger signs observed at admission showed increasing trends, where, over the years, children were more frequently presenting with difficulty breathing, dehydration, pallor, and neurological signs. Conversely, no trends were observed in the presentation of fever, cough, diarrhea, vomiting, rash, convulsion, jaundice or cerebral palsy (Fig 1D and S4 Fig, with detailed results in S3, S4 Tables).

Data presented as frequency (%) or median and interquartile range (IQR). €Missingness in anthropometry was undocumented in approximately 40% of health records, available numbers are specified for each measure and indicated per group. WHO AWaRe classification groups antibiotics into 'Access' (broad-spectrum with a lower risk of resistance, e.g., benzathine penicillin, amoxicillin and chloramphenicol) and 'Watch' (broad-spectrum with higher risk of selection for bacterial resistance, e.g., ceftriaxone, ciprofloxacin and meropenem). MUAC, mid-upper arm circumference; WAZ, weight-for-age z-score, HAZ, height-for-age z-score (if < 24 months, length-for-age z-score); WHZ, weight-for-height z-score (if < 24 months, length-for-age z-score).

### 10-year trends in HIV screening, vaccination status and antibiotic usage in children admitted with SAM

Across the 10-year span, 52% [95% CI, 50–55] of children had documented HIV results. While HIV testing significantly increased over the years, there was little to no rise in the detection of HIV reactivity (Fig 2A and S5 Table). Thus, although more children were being tested, the rates of HIV detection remained relatively stable in this cohort. The study also examined vaccination status. Complete vaccination was reported in 57% [95% CI, 55–60] of children, however, reports of complete vaccination significantly declined over the decade from 76% [95%CI 71–80] in 2011 to 27% [95%CI 20–34] in 2021 (Fig 2B and S6 Table). This decline coincided with an increase in unreported vaccination status of patients. Regarding antibiotic stewardship, the usage of 'Access' antibiotics declined over the 10-year period: 98% [95%CI 96–99] in 2011 to 84% [95%CI 76–90] in 2021 but the use of 'Watch'-classified antibiotics did not show a significant increasing trend ($p$ = 0.083) (Fig 2C and S7 Table).

**Table 1. Characteristics of children with severe acute malnutrition admitted to the Moyo NRU who died versus survived until discharge.**

| | Death n = 346 | Discharge n = 1,151 |
|---|---|---|
| **Characteristics** | | |
| Sex, female | 180/340 (53%) | 595/1133 (53%) |
| Age (months) | 18 (12, 29) | 24 (15, 35) |
| Breastfed (in children < 12 months) | 28/47 (60%) | 78/101 (77%) |
| **Nutritional status** | | |
| Edema, yes | 160/318 (50%) | 694/1114 (62%) |
| Edema severity | | |
| *Mild (+)* | 40/160 (25%) | 148/694 (21%) |
| *Moderate (++)* | 74/160 (46%) | 330/694 (48%) |
| *Severe (+++)* | 46/160 (29%) | 216/694 (31%) |
| MUAC$^\epsilon$ (cm) | 11.0 (10.0, 12.0), n = 170 | 11.8 (10.8, 13.0), n = 575 |
| *Wasting* | 10.0 (9.0, 11.0) | 11.0 (10.0, 11.5) |
| *Edema* | 11.8 (10.9, 13.0) | 12.9 (11.6, 14.0) |
| WHZ$^\epsilon$ | -3.9 (-4.9, -2.9), n = 178 | -2.8 (-3.9, -1.4), n = 656 |
| *Wasting* | -4.4 (-5.2, -3.5) | -3.6 (-4.6, -2.9) |
| *Eedema* | -3.1 (-4.1, -2.0) | -1.8 (-3.2, -0.66) |
| WAZ$^\epsilon$ | -4.6 (-5.4, -3.4), n = 296 | -3.6 (-4.8, -2.5), n = 1049 |
| *Wasting* | -4.8 (-5.8, -4.2) | -4.6 (-5.4, -3.7) |
| *Eedema* | -3.9 (-5.0, -2.9) | -3.1 (-4.1, -2.0) |
| HAZ$^\epsilon$ | -3.3 (-4.7, -2.3), n = 178 | -3.3 (-4.5, -2.2), n = 656 |
| **Underlying health status** | | |
| HIV reactivity | | |
| *Reactive* | 39 (11%) | 133 (12%) |
| *Exposed\* (< 18 months)* | 31 (9.0%) | 48 (4.2% |
| *Non-reactive* | 54 (16%) | 177 (15%) |
| *Untested* | 222 (64%) | 793 (69%) |
| Cerebral palsy | 29/234 (12%) | 112/772 (15%) |
| Vaccination | | |
| *Completed* | 198 (57%) | 654 (57%) |
| *Missed* | 46 (13%) | 132 (11%) |
| *Unrecorded* | 102 (29%) | 365 (32%) |
| **Symptoms at admission** | | |
| Dehydration | 89/273 (33%) | 152/945 (16%) |
| Fever (axillary temp ≥37.5˚C) | 165/301 (55%) | 551/992 (56%) |
| Convulsions | 5/88 (5.7%) | 13/304 (4.3%) |
| Diarrhea | 164/302 (54%) | 445/983 (45%) |
| Vomiting | 138/299 (46%) | 359/981 (37%) |
| Cough | 164/298 (55%) | 458/979 (47%) |
| Difficult breathing | 75/293 (26%) | 157/975 (16%) |
| Rash | 46/285 (16%) | 126/966 (13%) |
| Pallor | 33/287 (11%) | 103/961 (11%) |
| Jaundice | 13/285 (4.6%) | 30/958 (3.1%) |
| Neurological signs | 19/262 (7.3%) | 44/914 (4.8%) |
| **Antibiotics** | | |
| Access group | 327 (95%) | 1,080 (94%) |
| Watch group | 131 (38%) | 261 (23%) |

*(Continued)*

**Table 1.** (Continued)

|  | **Death**<br>**n = 346** | **Discharge**<br>**n = 1,151** |
|---|---|---|
| Number of antibiotics |  |  |
| *1 to 2* | 88 (25.1%) | 325 (29%) |
| *3* | 143 (41%) | 558 (48%) |
| *≥ 4* | 105 (30%) | 224 (19%) |
| **Outcome** |  |  |
| Time to death (days) | 3 (1, 7) | - |
| Time to discharge (days) | - | 7 (5, 9) |
| Readmission | 39 (11%) | 121 (11%) |

## 10-year trends indicate stable mortality rates and increasing readmissions in children hospitalized with SAM

Overall inpatient mortality across the 10-year period was 23% [95% CI, 21–25] (i.e. 346 mortality cases out of 1498 children) (Fig 2D and S8 Table) and the yearly mortality rate did not significantly decline over the 10 years, being 25% [95% CI, 20–31] in 2011 and 20% [95% CI, 14–27] in 2021 ($p$ = 0.23). However, the median time to mortality significantly decreased from 5.6 days [95% CI, 4.6–6.9] in 2011 to 3.5 days [95% CI, 2.5–4.7] in 2021 ($p$<0.01) (Fig 2E and

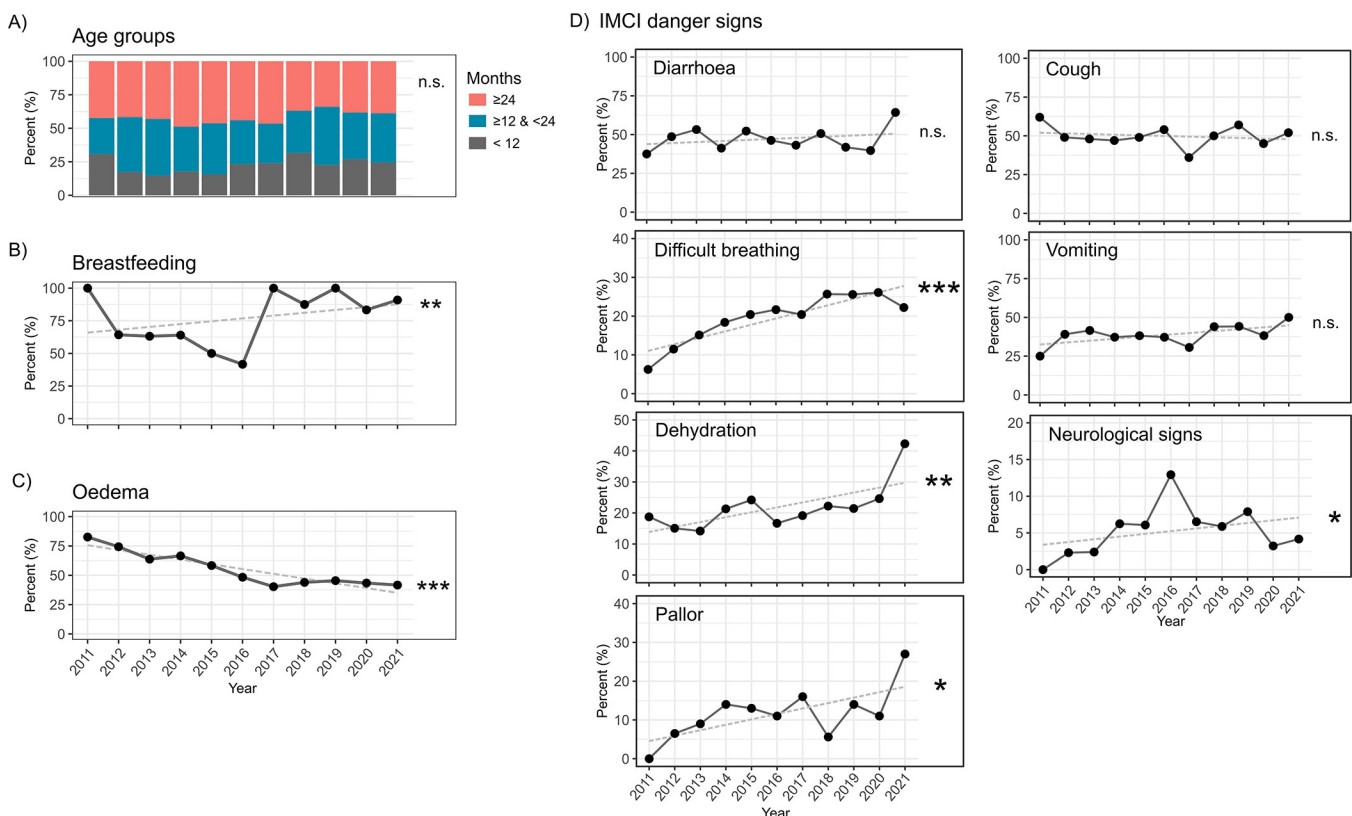

**Fig 1. Trends over 10 years (2011–2021) in clinical characteristics of children with severe acute malnutrition admitted at Moyo NRU.** Frequency per year of (A) Age groups as indicated by legend; (B) Reported breastfeeding in children less than 12 months of age; (C) Nutritional edema; and selected (D) Integrated Management of Childhood Illness (IMCI) danger signs. Linear and non-linear trends were tested with generalized additive models, full results are presented in S1, S3 Tables. Grey dashed line shows linear fit, and significance indicated at right: n.s., non-significant, *p<0.05, **p<0.01, ***p<0.001.

S8 Table). Also, while time to discharge decreased, the frequency of readmissions increased from 4.9% [95% CI, 3.3–7.4] in 2011 to 25% [95% CI, 18–33] in 2021 (*p*<0.0001).

## Clinical indicators associated with outcomes of mortality or readmission

We then evaluated which clinical features were most strongly related to mortality or readmission and whether these relationships changed over time. Logistic regression results are presented adjusted for age, sex and HIV status and additionally adjusted by WHZ (Table 2). Increased risk of mortality was associated with having lower age (i.e., < 2 years), very low WHZ (i.e., < -3.5 z-score), severe wasting (as opposed to edema), dehydration, diarrhea, vomiting, cough, and difficulty breathing. With additional adjustment for WHZ, the clinical signs that remained associated with mortality were dehydration and vomiting. Apart from age and WHZ, dehydration showed the strongest association with mortality, where 37% [95% CI, 31–43] of children with dehydration died compared to 19% [95% CI, 16–21] in children without. The relationships between these clinical signs and mortality did not change over the 10-year period, showing no interaction with time. Due to missingness and reduction in sample size, overall multivariate models were not considered.

Hospital readmission was more common in children with severe wasting (15% [95% CI, 12–19] vs. 7% [95% CI, 6–10] in children with nutritional edema; OR, 0.44 [0.30, 0.65], *p*<0.001) (S9 Table). While HIV reactivity was not associated with mortality, it was linked to increased readmission (9.7% [8.1, 12] vs. 17% [11, 24]; OR, 1.8 [95%CI, 1.0–3.0], *p* = 0.026).

Competitive risk analysis was used to further evaluate the clinical features strongly associated with mortality to assess differences in the daily risk of death or discharge (Fig 3 and S10, S11 Tables). This revealed that children with both very low WHZ and lower age (Fig 3A), or

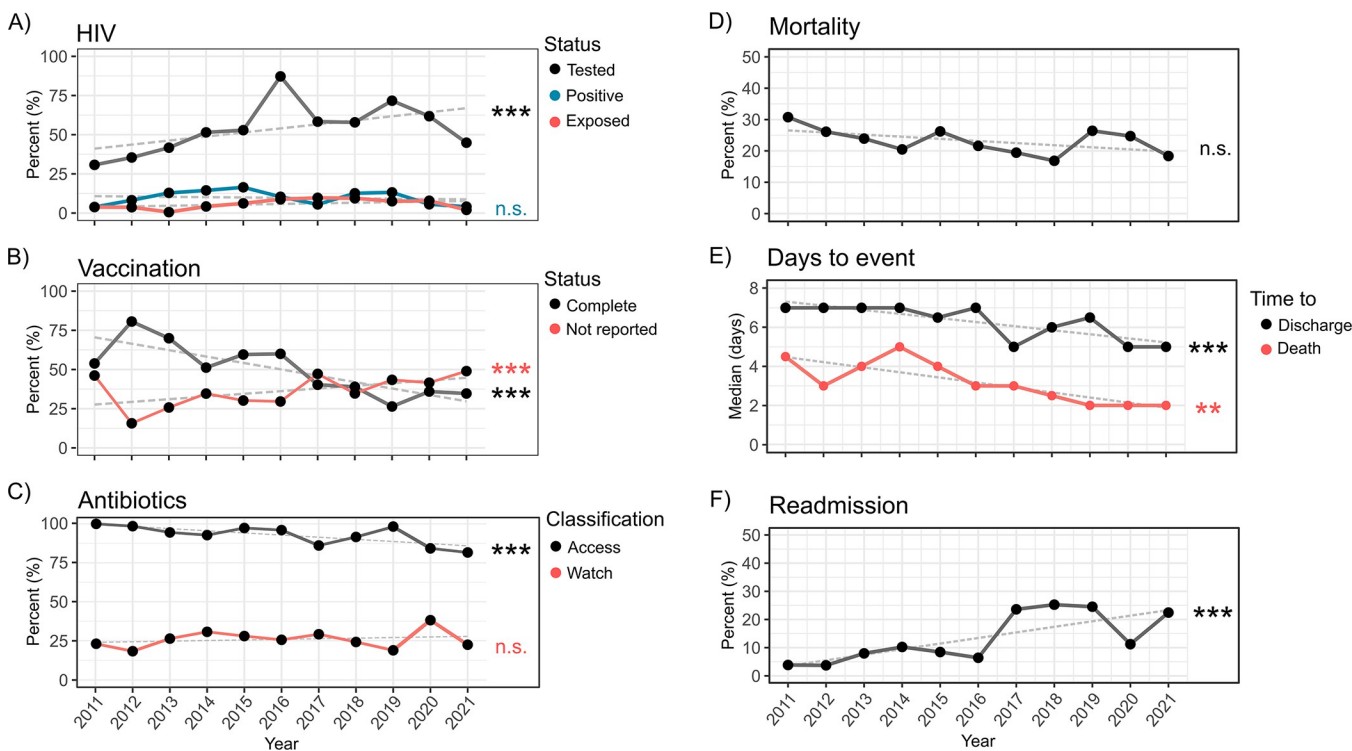

**Fig 2. Trends over 10 years in children with severe acute malnutrition admitted at Moyo NRU.** (A) HIV status as per legend; (B) Vaccination status (black line) and unreported vaccination (red); (C) Antibiotic usage as per WHO-Aware classification (Access, black line; and Watch, red); (D) Mortality; (E) Days to discharge (black line) or to death (red); and (F) Readmission. Linear and non-linear trends tested with general additive models, full results presented in S5–S8 Tables. Grey dashed line shows linear fit, and significance indicated at right: n.s., non-significant, *p<0.05, **p<0.01, ***p<0.001.

**Table 2. Clinical features associated with mortality of children with severe acute malnutrition admitted to the Moyo NRU across the 10-year period.**

| Clinical feature | Mortality % [95% CI][1] | OR [95% CI] adjusted[2] | p | OR [95% CI] adjusted[3] | p |
|---|---|---|---|---|---|
| **WHZ** | | | | | |
| ≥ -3.5 | 15 [12, 18] | Ref | | - | |
| < -3.5 | 32 [27, 37] | 2.76 [1.95, 3.95] | *** | - | |
| **Age** | | | | | |
| ≥ 5y | 16 [11, 22] | Ref | | Ref | |
| ≥ 2 & < 5 years | 19 [16, 23] | 1.29 [0.82, 2.08] | | 5.46 [2.23, 16.53] | *** |
| < 2 years | 27 [24, 31] | 2.06 [1.34, 3.26] | ** | 6.99 [2.95, 20.69] | *** |
| **Eedema** | | | | | |
| No | 27 [24, 31] | Ref | | Ref | |
| Yes | 19 [16, 22] | 0.63 [0.49, 0.82] | *** | 1.45 [0.97, 2.18] | 0.070 |
| **Dehydration** | | | | | |
| No | 19 [16, 21] | Ref | | Ref | |
| Yes | 37 [31, 43] | 2.39 [1.75, 3.26] | *** | 1.64 [1.04, 2.57] | 0.033 |
| **Diarrhea** | | | | | |
| No | 20 [18, 24] | Ref | | Ref | |
| Yes | 27 [24, 31] | 1.39 [1.07, 1.81] | 0.013 | 1.07 [0.73, 1.57] | n.s. |
| **Vomit** | | | | | |
| No | 21 [18, 24] | Ref | | Ref | |
| Yes | 28 [24, 32] | 1.5 [1.15, 1.96] | ** | 1.57 [1.07, 2.3] | 0.021 |
| **Cough** | | | | | |
| No | 20 [18, 24] | Ref | | Ref | |
| Yes | 26 [23, 30] | 1.35 [1.03, 1.76] | 0.027 | 1.13 [0.77, 1.67] | n.s. |
| **Difficult breathing** | | | | | |
| No | 21 [19, 24] | Ref | | Ref | |
| Yes | 32 [27, 39] | 1.77 [1.28, 2.42] | *** | 1.11 [0.68, 1.78] | n.s. |
| **HIV** | | | | | |
| No | 23 [21, 25] | Ref | | Ref | |
| Yes | 25 [19, 32] | 1.11 [0.75, 1.63] | n.s. | 0.91 [0.48, 1.64] | n.s. |

[1]Percentage and 95% confidence intervals (95% CI) of mortality estimated for each group classification derived from unadjusted logistic regression models. Odds ratio (OR) and 95%CI are presented for

[2]models adjusted for age, sex, and HIV status and

[3]models additionally adjusted for WHZ. WHZ, weight-for-height/length z-score.

both diarrhea and dehydration (Fig 3B) had an increased daily risk of death and a decreased probability of discharge. This underscores the interplay between these clinical features. Also, cough, difficulty breathing, and vomiting were associated with increased risk of mortality and decreased probability of discharge (Fig 3C–3F). The interaction between vomiting and dehydration was not significant, but 90% (218/241) of dehydration cases co-occurred with symptoms of gastroenteritis (i.e., vomiting, diarrhea, or both) (S5 Fig). Of 326 children who had very low WHZ, 180 (55%) also had dehydration, diarrhea, vomiting or a combination of these conditions (S5 Fig).

## Discussion

This retrospective analysis of children with severe wasting and/or nutritional edema admitted to the largest NRU in Malawi shows that between 2011 and 2021 patient mortality risk remained persistently high (23%). In 2021, deaths occurred approximately 3 days earlier than

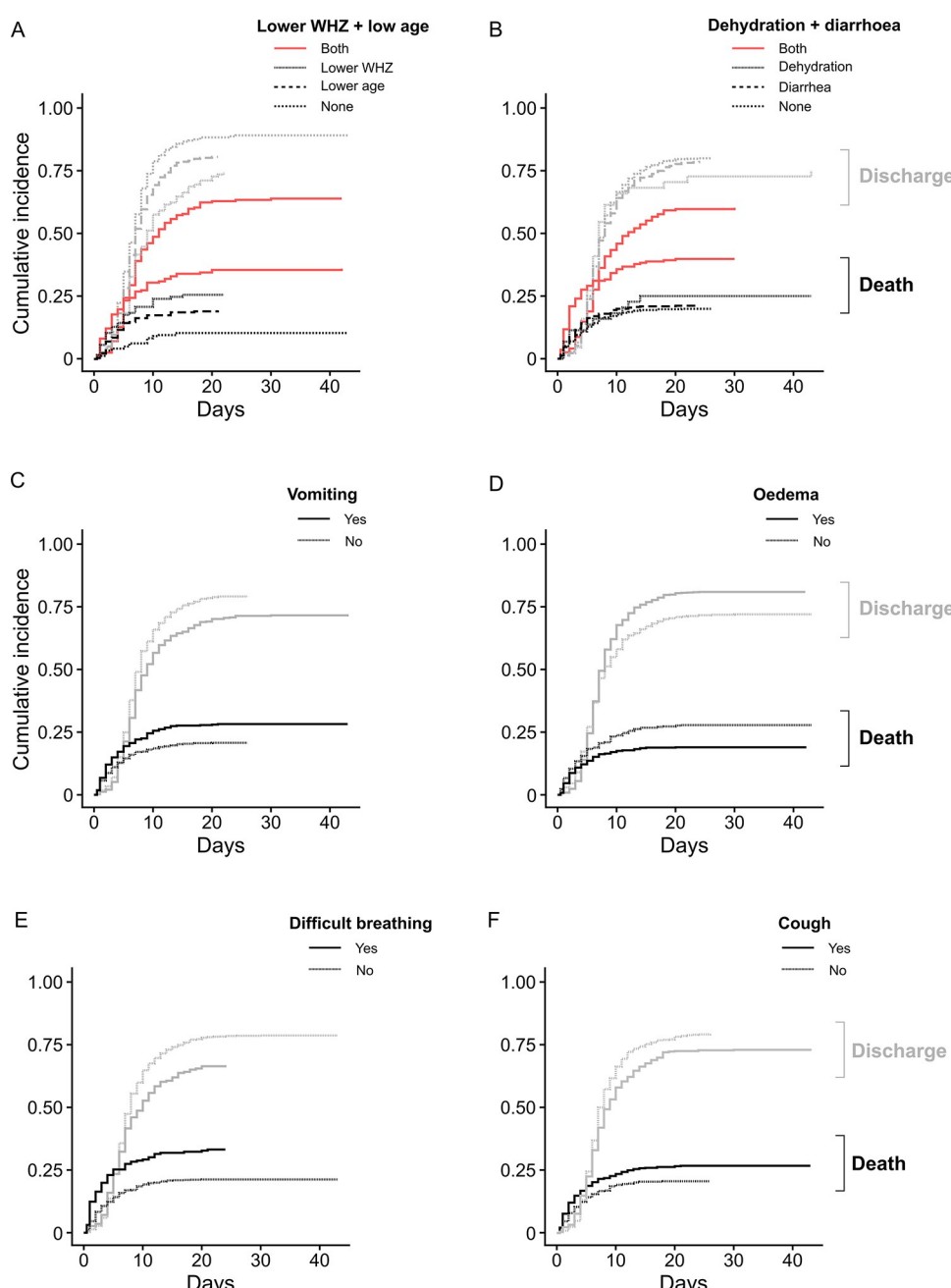

**Fig 3. Differences in the daily probability of mortality or discharge associated with selected clinical features at admission in children admitted to the Moyo NRU over the 10-year period.** Estimated cumulative incidence curves represent the probability of death (in black) or discharge (in grey) at any given day of hospitalization depending on having (A) either lower WHZ (i.e., WHZ < -3.5 z-score) or lower age (i.e., < 2 years) or both (indicated in red); (B) only diarrhea or dehydration, or both (indicated in red); (C) vomiting; (D) edema; (E) difficulty breathing; or (F) cough, as indicated by legends. Group differences were tested using competitive risk analysis which showed that both the rate of mortality and of discharge differed depending on the presence of these clinical features (Results detailed in S10, S11 Tables). WHZ, weight-for-height/length z-score.

in 2011. Over the decade, several IMCI danger signs and other clinical symptoms were more frequently reported at hospital admission in children with SAM. However, there was a significant 40% reduction in the prevalence of nutritional edema. Several clinical factors identified at

admission were associated with mortality including young age and low WHZ, signs of dehydration and diarrhea/vomiting, or respiratory distress. HIV-reactivity was associated with an increased risk for readmission.

To the best of our knowledge, this is the first study to highlight changing clinical trends over a 10-year period of child admissions to a nutritional rehabilitation unit in low resource settings. While changing trends in anthropometry were not found (potentially due to missingness and lack of power), we found that indicators of dehydration, respiratory distress, pallor, and neurological signs were more prevalent in 2021 compared to a decade earlier. This aligns with our clinical experience, which suggests that while fewer children are admitted to the Moyo NRU [30], those that are present with a higher disease intensity, an effect likely derived from well-functioning referral processes. Thus, the change in clinical presentation could be explained by QECH receiving, by design, the more severe cases which likely contributes to the persistent high in-hospital mortality rates. Also, recent data indicates that nearly 50% of child deaths linked to acute illnesses occur after discharge [4]. Given the shortened length of hospital stay and lack of post-discharge data, mortality rates are likely underestimated in these children with high disease intensity.

The 20% mortality rate observed in 2021 at Moyo NRU, the largest facility of its kind in Malawi, still stands notably higher than the national mortality rate of 11.6% reported in 2018 by other NRUs in the country [31]. However, our findings align with a comparable large tertiary hospital in Uganda, where a mortality rate of 25% was observed among acutely ill children with SAM [32]. Moyo NRU serves as a referral center for the entire Southern Region of Malawi (population approximately 8 million), extending its inpatient treatment services to both urban and rural districts of Blantyre, encompassing a population of roughly 1 million. Many children are directed to Moyo after failing to respond to nutritional interventions provided in primary and secondary health facilities. From a broader perspective, the persistently high mortality rate of Moyo NRU can be due to successful referral systems where the most ill children are effectively redirected to facilities with higher capacity.

We noted a consistent decrease in children admitted with nutritional edema over the decade, a trend also described in outpatient treatment programs in Malawi [30]. The overall prevalence of nutritional edema could be lower, possibly due to the adoption of community management of acute malnutrition (CMAM) national guidelines [24]. Implemented by 2012 across over 90% percent of health facilities in Malawi, these guidelines recommended that children with mild to moderate nutritional edema but with good appetite and no medical complications be managed as outpatients. Although there is no direct evidence of the impact of CMAM on mortality in Malawi or similar settings, there has been some evidence of its positive effect on primary outcomes including recovery and default rates of children with SAM [33]. While historically children with nutritional edema had a higher mortality risk [34], we found these children to have approximately 10% lower risk compared to those with severe wasting. However, edema was recently associated with an increased risk of mortality in Malawi and Kenya [11]. Thus, the association between edema and mortality has been conflicting, especially with the emergence of certain co-morbidities such as HIV [5, 35].

Dehydration in the presence of diarrhea emerged as a significant factor associated with mortality, a finding also described in other studies [8, 36–39]. Recognizing dehydration can be challenging in children with SAM as typical physical indicators can be misinterpreted and accurate recognition often depends on the expertise of healthcare workers [36, 40]. Thus, groups have advocated for enhanced training to improve the detection of dehydration in children with severe wasting or nutritional edema [19, 36]. Our previous investigation in Malawi and Kenya (done from 2014 to 2015) found an association between mortality risk and low anthropometry, reduced consciousness, respiratory distress, poor feeding, and diarrhea but

dehydration was not investigated [11]. While updated WHO guidance recommends fluid management procedures to treat dehydration [41], the quality of evidence is low [42], thus, further research is needed to better guide clinical practice.

Over the decade, readmissions increased which could be indicative of the persistent vulnerability of these children after discharge and may also relate to the shortened hospital stay observed. Recently, 2 large-scale studies in LMICs showed that almost half of all deaths of hospitalized children occurred following discharge, with post-discharge mortality around 5% and deaths often happening at home [4, 13]. Children at risk for readmission could be identified using simple IMCI criteria and readmission may be as predictable as in-patient or post-discharge mortality [13]. Our study supports the conclusions of the CHAIN network cohort study [4], emphasizing the need to shift towards a child-centred, risk-based approach to further reduce mortality of vulnerable, undernourished children during index admission, readmission and post-discharge.

Strengths of our analysis are the longitudinal 10-year time frame, the large number of children studied, and the broad data collection covering both clinical profiles and outcomes relevant to health system implementation. However, we do acknowledge several limitations. Our dataset included clinical warning signs at admission only. Laboratory findings were often poorly archived (e.g., thermal paper print outs with faded ink) and radiological assessments were not systematically done or archived (e.g., for tuberculosis). Thus, we were unable to capture laboratory or radiological findings or include rarer chronic conditions associated with malnutrition like tuberculosis. We used archived paper-based medical records originally intended for patient care rather than research. The main limitation of the study is the high missingness rates in anthropometric measurements. While we made diligent efforts to assess and account for missing data, the potential for reporting biases exist, e.g., healthcare providers possibly avoiding performing anthropometric measurements in sicker children, or the decrease in reporting of vaccination status. The review of archived records can also be useful in identifying lacks in clinical documentation that can be addressed, improving data capture would be valuable for both research and clinical practice. Lastly, we encountered challenges relating to clarity of documentation in older records, primarily due less structured case report forms and factors like ink fading, an issue relevant to any long-term archiving systems in LMICs.

## Conclusions

In conclusion, this study analyzed the medical records of children admitted to the largest NRU in Malawi over a decade (2011–2021). Cases of nutritional edema declined significantly which may relate to the implementation of CMAM programs. While mortality was persistently high despite aligning with WHO guidelines for the treatment of SAM, the disease intensity of admitted children increased over the 10-year span. Heightened risk of mortality was associated with younger children with a low WHZ, signs of dehydration and diarrhea/vomiting, and respiratory distress. Together this underscores the urgent need to transition from a one-size-fits-all approach to a child-centered, risk-based strategy that accounts for specific combinations of symptoms (e.g., fluid management strategies in children with edema vs. with both diarrhea and dehydration). It would be important to periodically review criteria for hospitalization and discharge and allocate resources considering the changing profiles of clinical presentation that may predict outcomes differently over time. Prioritizing tailored interventions including improved and ongoing risk stratification, offering clear access to post-discharge care by, for example, implementing digital health technologies for follow-up care and reducing barriers to readmission in resource-constrained settings like Malawi holds the potential to reduce mortality.

## Supporting information

**S1 Fig. Study flow chart representing selection procedure of medical files of severely malnourished children admitted at Moyo Rehabilitation Unit between 2011 and 2021.** From a random start point (selected between 1 and 3), every 3rd file was selected for inclusion. One duplicated file was removed leaving 1,497 medical files included in the final analysis.
(PDF)

**S2 Fig. Trends over 10 years in severity of edema in children admitted at Moyo Rehabilitation Unit with nutritional edema.** Frequency of edema (+), (++), (+++) as defined by the World Health Organisation presented across years by solid lines and black dots coloured as per legend. Linear and non-linear trends tested with general additive models. Grey dashed lines indicate linear fit with significance at right: n.s., non-significant, *p<0.05, **p<0.01, ***p<0.001.
(PDF)

**S3 Fig. Anthropometry and growth metrics of children admitted with severe wasting and/or nutritional edema at MOYO NRU over the 10-year period.** A) HAZ across all children; B-D) MUAC, WHZ and WAZ in children with severe wasting; E-G) MUAC, WHZ and WAZ in children with nutritional edema. Solid midline (black) with dots shows median of growth metrics across years, top and bottom solid lines (red) are the high and low interquartile range. Linear and non-linear trends tested with general additive models. Grey dashed lines indicate linear fit with significance at right: n.s., non-significant, *p<0.05, **p<0.01, ***p<0.001. MUAC, mid upper arm circumference; WAZ, weight-for-age z-score, HAZ, height-for-age z-score (if < 24 months, length-for-age z-score); WHZ, weight-for-height z-score (if < 24 months, length-for-age z-score).
(PDF)

**S4 Fig. Trends over 10 years in the Integrated Management of Childhood Illness (IMCI) danger signs presented by children with severe wasting and/or nutritional edema admitted at Moyo NRU.** Frequency of A) Fever, B) Convulsion, C) Cerebral palsy, D) Rash, E) Jaundice. Linear and non-linear trends tested with general additive models. Grey dashed lines indicate linear fit with significance at right: n.s., non-significant, *p<0.05, **p<0.01, ***p<0.001.
(PDF)

**S5 Fig. Co-occurrence of symptoms related to gastroenteritis in children admitted at Moyo Rehabilitation Unit.** A) Dehydration, diarrhorea, and vomiting; and B) Dehydration, diarrhoea, vomiting very low WHZ (< -3.5 z-score).
(PDF)

**S1 Table. Trends in age, sex, and breastfeeding over the 10-year period in children with severe wasting and/or nutritional edema admitted to Moyo NRU.** Median and interquartile range or n (%) presented as appropriate. Prevalence of breastfeeding was calculated in children younger than 12 months of age. Linear and non-linear trends were tested with general additive models.
(PDF)

**S2 Table. Trends in nutritional edema and severity of edema over the 10-year period.** n (%) present severity of edema (+), (++), (+++) as defined by the World Health Organisation. Linear and non-linear trends were tested with general additive models.
(PDF)

**S3 Table. Trend in selected Integrated Management of Childhood Illness (IMCI) danger signs for prevalence of diarrhea, difficult breathing, dehydration, and pallor over the 10-year period in children with severe wasting and/or nutritional edema admitted to Moyo NRU.** n (%) presented. Linear and non-linear trends were tested with general additive models. (PDF)

**S4 Table. Trend in selected Integrated Management of Childhood Illness (IMCI) danger signs for prevalence of cough, vomitting, and neurological signs over the 10-year period in children with severe wasting and/or nutritional edema admitted to Moyo NRU.** n (%) presented. Linear and non-linear trends were tested with general additive models. (PDF)

**S5 Table. Trend in HIV reactivity and testing over the 10-year period in children with severe wasting and/or nutritional edema admitted to Moyo NRU.** n (%) present frequency of HIV. Linear and non-linear trends were tested with general additive models. (PDF)

**S6 Table. Trend in vaccination reporting over the 10-year period in children with severe wasting and/or nutritional edema admitted to Moyo NRU.** Frequencies presented as n(%). Linear and non-linear trends were tested with general additive models. (PDF)

**S7 Table. Trends in antibiotic usage over the 10-year period in children with severe wasting and/or nutritional edema admitted to Moyo NRU.** Frequencies presented as n(%). Linear and non-linear trends were tested with general additive models. Antibiotic classification as per WHO-Aware framework relating to potential of contributing to antibiotic resistance. WHO AWaRe classification groups antibiotics into 'Access' (broad spectrum antibiotics with a lower risk of resistance, e.g., benzathine penicillin, amoxicillin and chloramphenicol) and 'Watch' (broad spectrum with higher risk of selection of bacterial resistance, e.g., ceftriaxone, ciprofloxacin and meropenem). (PDF)

**S8 Table. Trends in mortality, readmission and time to death or discharge over the 10-year period in children with severe wasting and/or nutritional edema admitted to Moyo NRU.** Median and interquartile range or n (%) presented as appropriate. Linear and non-linear trends were tested with general additive models. (PDF)

**S9 Table. Clinical features associated with readmission of children with severe wasting and/or nutritional edema admitted to MOYO nutritional rehabilitation unit across the 10-year period.** Results from logistic regression analysis presenting odds ratios (OR) and 95% confidence intervals (95%CI). WHZ, weight-for-height/length z-score. (PDF)

**S10 Table. Clinical features associated with daily probability of mortality in children with severe wasting and/or nutritional edema admitted to MOYO nutritional rehabilitation unit across the 10-year period.** Results from competitive risk analysis presenting unadjusted or age- and WHZ- adjusted daily risk of mortality as odds ratios (OR) and 95% confidence intervals (95%CI). WHZ, weight-for-height/length z-score. (PDF)

**S11 Table. Clinical features associated with daily probability of discharge in children with severe wasting and/or nutritional edema admitted to MOYO nutritional rehabilitation**

**unit across the 10-year period.** Results from competitive risk analysis presenting unadjusted or age- and WHZ- adjusted daily risk of discharge as odds ratios (OR) and 95% confidence intervals (95%CI). WHZ, weight-for-height/length z-score.
(PDF)

**S1 Dataset.**
(CSV)

## Acknowledgments

We acknowledge the work, dedication and commitment of all nursing staff and medical personnel at Moyo NRU over the years and are grateful for their commitment to patient care. We are also thankful for the contribution of Christabelle Potani whom diligently worked to enter data.

## Author Contributions

**Conceptualization:** Mphatso Nancy Chisala, Celine Bourdon, Emmanuel Chimwezi, Allison I. Daniel, Dominic Wang, Linnea Weise, Isabel Potani, Robert J. H. Bandsma, Wieger P. Voskuijl.

**Data curation:** Mphatso Nancy Chisala, Celine Bourdon, Emmanuel Chimwezi, Allison I. Daniel, Chikondi Makwinja, Dominic Wang, Linnea Weise, Wieger P. Voskuijl.

**Formal analysis:** Mphatso Nancy Chisala, Celine Bourdon, Allison I. Daniel.

**Funding acquisition:** Mphatso Nancy Chisala.

**Investigation:** Mphatso Nancy Chisala, Emmanuel Chimwezi, Dominic Wang, Wieger P. Voskuijl.

**Methodology:** Mphatso Nancy Chisala, Celine Bourdon, Emmanuel Chimwezi, Dominic Wang, Linnea Weise, Isabel Potani, Wieger P. Voskuijl.

**Project administration:** Celine Bourdon, Emmanuel Chimwezi, Chikondi Makwinja, Isabel Potani, Emmie Mbale, Wieger P. Voskuijl.

**Resources:** Robert J. H. Bandsma, Wieger P. Voskuijl.

**Software:** Mphatso Nancy Chisala, Celine Bourdon, Emmanuel Chimwezi, Chikondi Makwinja, Dominic Wang.

**Supervision:** Celine Bourdon, Emmanuel Chimwezi, Chikondi Makwinja, Isabel Potani, Emmie Mbale, Robert J. H. Bandsma, Wieger P. Voskuijl.

**Validation:** Celine Bourdon, Emmanuel Chimwezi, Chikondi Makwinja, Wieger P. Voskuijl.

**Visualization:** Mphatso Nancy Chisala, Celine Bourdon, Wieger P. Voskuijl.

**Writing – original draft:** Mphatso Nancy Chisala, Celine Bourdon, Allison I. Daniel, Chikondi Makwinja, Robert J. H. Bandsma, Wieger P. Voskuijl.

**Writing – review & editing:** Mphatso Nancy Chisala, Celine Bourdon, Emmanuel Chimwezi, Allison I. Daniel, Chikondi Makwinja, Dominic Wang, Linnea Weise, Isabel Potani, Emmie Mbale, Robert J. H. Bandsma, Wieger P. Voskuijl.

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
