## [Decision Letter · Decision Letter 0]

22 Aug 2024

PONE-D-24-21799Ten-year trends in clinical characteristics and outcome of children hospitalized with severe wasting or nutritional oedema in Malawi (2011-2021): Declining admissions but worsened clinical profilesPLOS ONE

Dear Dr. Bourdon,

Thank you for submitting your manuscript to PLOS ONE. After careful consideration, we feel that it has merit but does not fully meet PLOS ONE’s publication criteria as it currently stands. Therefore, we invite you to submit a revised version of the manuscript that addresses the points raised during the review process.

We look forward to receiving your revised manuscript.

Kind regards,

Nigusie Selomon Tibebu, MSc

Academic Editor

PLOS ONE

Additional Editor Comments (if provided):

Reviewers' comments:

Reviewer's Responses to Questions

**Comments to the Author**

1. Is the manuscript technically sound, and do the data support the conclusions?

Reviewer #1: Partly

Reviewer #2: Yes

2. Has the statistical analysis been performed appropriately and rigorously? 

Reviewer #1: No

Reviewer #2: Yes

3. Have the authors made all data underlying the findings in their manuscript fully available?

Reviewer #1: No

Reviewer #2: Yes

4. Is the manuscript presented in an intelligible fashion and written in standard English?

Reviewer #1: Yes

Reviewer #2: Yes

5. Review Comments to the Author

Reviewer #1: Re: Ten-year trends in clinical characteristics and outcome of children hospitalized with

severe wasting or nutritional oedema in Malawi (2011-2021): Declining admissions but

worsened clinical profiles

Overall comments

There should be a flow chart illustrating patients recruited/enrolment as appropriate

Abstract “

Results-line 45 mentioned anthropometry-indicate which parameters you assessed.

Line 47 “HIV testing increased but detection remained at 11%.” Is this annually or overall? Clarify

Introduction

Methods

“Medical files were accessed from 8th of June 2020, to 20th of May, 2022” Is this for retrieval of the files?

“Files were stored by date of admission, thus, starting from a randomly selected file, every third record was marked for inclusion” please provide ref/any justification for your decision.

“Research staff” how many?

“demographics (i.e., age, sex, breastfeeding in children < 12 months)” Breastfeeding is not demographics and can it described what you meant breastfeeding less than 12? Still breasting/EBM

“‘Access’ (broad-spectrum with a lower risk of resistance) and ‘Watch’ (broad-spectrum with higher risk of selection for bacterial resistance)” Provide examples of these antibiotics from cases.

“Implausible anthropometric values of WHZ, HAZ and WAZ were filtered if exceeded +4 or fell below -10 z-scores” how many ? provide the n

Results

“From 2011 to 2021, Moyo NRU had 4494 admissions, of which 1498 (33%) were randomly selected for inclusion in this retrospective analysis” A bit difficult to understand while the number sampled was not more than this considering the high number of missing variables and incomplete data? Kindly provide rationale for this?

Table 1; Insert exact number of available anthropometry rather indicating missing

[MUAC, WHZ, WAZ, HAZ], What the access and watch group of antibiotics with examples.

“Missingness in anthropometry was undocumented in approximately 40% of health records” this is the greatest limitation of this study and is missing the study limitation, kindly provide exact number that are using.

“This decline coincided with an increase in unreported vaccination status of patients” I can not locate this in the results?

(i.e. 346 mortality cases out of 1498 children)-This appear incorrect since close to 40% of children did not have anthropometry for proper classification/inclusion/exclusion

Table 2-can you clarify the models, model 2 did not include anthropometry???

Discussion

“However, there was a significant 40% reduction in the prevalence of nutritional edema” insert n before the percent ?

“To the best of our knowledge, this is the first study to cover a 10-year period and highlight changing trends in clinical presentation” This statement should be re-casted, this data has close to half missing with no details of what is missing in the anthropometry

“We were unable to capture laboratory or radiological findings or include specific chronic conditions associated with malnutrition like tuberculosis due to low recorded prevalence” How low is this? This should have been reported, it is an important co-morbidity that may influence outcomes

Figures 1 to 4 are poor qualities and hard to read, kindly revise to enhance readability

S1 Table: The age and sex has non-linear trend, what inform the choice to report non-linear vs linear trend

S2 Table: Again non-linear trend was not reported for oedema grade 2 and 3, even if it is not significant, this should be reported for consistency and if it is not done, a footnote should be added indicating why it was not done.

S3 Table; Same comments as S2 Table: Non-linear trend not reported for some variables?

S7 Table: Same comments

S8 Table: ? Non-linear trends? Also, the number of some variables are small? 26 for a whole year?

Reviewer #2: Methods

Handling of Missing Data: While you mention evaluating trends in missing values and also mention missingness as a contributing limitation towards your choice of methodology (lines 214-215), It is important to detail how missing data were handled in the analysis (e.g., complete case analysis, imputation methods).

It is not clear if you only included single entry of each child as a criterion or if it is possible that in the 10 year period a single child could have been admitted several times. If this is the case, it is not clear as to how you dealt with the repeated measures within subjects: If the same children appear multiple times in the dataset, you should include random effects for the child to account for intra-child correlation. This introduces non-independence of the observations. If this is indeed the case, then perhaps mixed effects GAMs would be appropriate.

Discussion

The synthesis of the evidence presented in the statement could be considered misleading. This sentence should be improved to avoid misinterpretation. line 258-259 "This aligns with our clinical experience, which suggests that while fewer children are admitted to the Moyo NRU,(28) they present with a higher disease intensity." Is this not expected?? This is a tertiary (referral centre) where referrals are made based on disease intensity I presume. Therefore it is inherently predisposed to receiving more severe cases based on criteria. If fewer children are being admitted, the proportion of severe cases might naturally appear higher because only the most severe cases are referred.

line 293 "Over the decade, readmissions increased which could be indicative of the persistent vulnerability of these children after discharge and may also relate to the shortened hospital stay observed."

I note that readmissions are mentioned here bringing back to how intra-child correlations were addressed.

6. PLOS authors have the option to publish the peer review history of their article (what does this mean?). If published, this will include your full peer review and any attached files.

Reviewer #1: No

Reviewer #2: No

---

## [Author Response · Author response to Decision Letter 0]

17 Sep 2024

Ten-year trends in clinical characteristics and outcome of children hospitalized with severe wasting or nutritional oedema in Malawi (2011-2021): Declining admissions but worsened clinical profiles.

Reviewer #1: 

Overall 

1) There should be a flow chart illustrating patients recruited/enrolment as appropriate.

Response: Our target population consisted of all archived files of admissions at Moyo between 2011 and 2021 (N=4,494). We employed a simple selection method, choosing every third file from a random starting point. This identified 1,498 files, of which 1 was a duplicated record and was removed. Thus, 1,497 medical files were included for analysis. 

We agree that it is standard practice to include a flow chart, therefore, one was added as a supplementary figure (S1 Fig) and we have clarified the selection process in methods (line 98-102), results (line 163-165) and figure legend of S1.

Abstract

2) Results-line 45 mentioned anthropometry-indicate which parameters you assessed.

Response: We have clarified the anthropometric parameters assessed. The section now reads as follows: “Median age at admission (23 months, IQR 14, 34) or anthropometry (i.e., weight-for-age, height-for-age and weight-for-height) did not change over the 10-years.” See abstract line 45 to 46.

3) Line 47 “HIV testing increased but detection remained at 11%.” Is this annually or overall? Clarify. 

Response: We clarified that this refers to the annual HIV testing and positive detection rate. The sentence now reads as follows: “Yearly HIV testing increased but positive-detection remained around 11%”. See line 48.

Introduction

No comments

Methods

4) Medical files were accessed from 8th of June 2020, to 20th of May, 2022” Is this for retrieval of the files?

Response: These dates are referring to when the files were retrieved for data entry. This is better clarified as follows: “Medical files were retrieved for data entry from 8th of June, 2020, to 20th of May, 2022.” See line 98-99. 

5) Files were stored by date of admission, thus, starting from a randomly selected file, every third record was marked for inclusion” please provide ref/any justification for your decision.

Response: We sampled every kth record (here, k being equal to 3). This is a common method of sampling patient records retroactively. 

“The third sampling technique is referred to as systematic sampling. Using this procedure, the researcher selects every k-th medical record for coding.” From Matt, V. and Matthew, H., J Educ Eval Health Prof. 2013; 10: 12). 

However, we added a layer of randomness by not starting with the first entry available but from a randomly selected start position between the numbers 1-3. This allowed us to sample across the 10-year span and calculate a number of records that could be feasibly entered considering resources. We have added the reference and better described the systematic sampling approach in the Methods selection, text as follows: “We followed a systematic sampling technique to select the paper-based records.(25) From files organized by date, a random start point (selected between 1 and 3) was indicated by the data manager. From that point, every 3rd file was flagged for inclusion which were then entered over time by 4 research staff into a customized database built with COMMCARE (https://www.dimagi.com/commcare/features/), an open-access digital data entry platform.” See line 99-103.

6) “Research staff” how many? 

Response: Overtime, 4 different research staff entered data for the study (E.C., C.M., C.B., C.P.). The data manager E.C. oversaw the full process throughout data collection and data entry took place in his office to facilitate guidance. The number of research staff that entered data is now mentioned. See line 102.

7) “demographics (i.e., age, sex, breastfeeding in children < 12 months)” Breastfeeding is not demographics and can it described what you meant breastfeeding less than 12? 

Response: Thank you for the correction. The sentence now reads as follows: “De-identified information included: demographics (i.e., age, sex), breastfeeding and anthropometry (i.e., height, weight, and mid-upper arm circumference (MUAC)).” See line 105-106.

Based on this comment we realised that the age-indicator “breastfeeding in children < 12 months” is not relevant here. Breastfeeding yes/no was evaluated for all children, but the prevalence of reported breastfeeding was analysed in children under 12 months of age (i.e., in those most likely to still be breastfeeding). Therefore, we moved the age specification to the description of the analysis as follows: “Report of breastfeeding was evaluated in children younger than 12 months of age.” See line 130-131.

We also changed the heading of the Table 1 from “Demographics” to “Characteristics” and the first header in the result section in consideration of this comment. 

8) “Access” (broad-spectrum with a lower risk of resistance) and “Watch” (broad-spectrum with higher risk of selection for bacterial resistance). Provide examples of these antibiotics from cases.” 

Response: Examples are now provided, text reads as follows: “We also assessed how antibiotic use changed based on updated WHO AWaRe classification which groups antibiotics into ‘Access’ (broad-spectrum with a lower risk of resistance, e.g., benzathine penicillin, amoxicillin and chloramphenicol) and ‘Watch’ (broad-spectrum with higher risk of selection for bacterial resistance, e.g., ceftriaxone, ciprofloxacin and meropenem).” See line 118-121. 

9) “Implausible anthropometric values of WHZ, HAZ and WAZ were filtered if exceeded +4 or fell below -10 z-scores” how many ? provide the n; 

Response: The numbers have been included. The sentence reads as follows: “Implausible anthropometric values of height (<30 cm and age > 12 months, n=7), weight (> 40 kg, age < 24 months, n=5) were filtered, and z-scores of >4 or <-10 were also removed (WHZ, n=4; HAZ, n=8; WAZ, n=1).” See line 127-129.

Results

10) “From 2011 to 2021, Moyo NRU had 4494 admissions, of which 1498 (33%) were randomly selected for inclusion in this retrospective analysis” A bit difficult to understand while the number sampled was not more than this considering the high number of missing variables and incomplete data? Kindly provide rationale for this?

Response: As mentioned above, we sampled every kth record, a common method of systematically sampling patient records (Matt, V. and Matthew, H., J Educ Eval Health Prof. 2013; 10: 12). We also clarified that we added a layer of randomness by not starting with the first entry available but from a randomly selected start position between the numbers 1-3. This allowed sampling across the 10-year span and to include a feasible number of records. Each record took on average 30-40 minutes to be entered, and staff aimed to enter 5-6 records in the morning and 3-4 in the afternoon which is a significant amount of manpower to complete entry for 1,498 files. This also guided the decision to choose the systematic sub-sampling technique of every kth record. 

We agree with the reviewer that the sentence was unclear and has been edited as follows: “From 2011 to 2021, Moyo NRU had 4,494 admissions. From these, we systematically sub-sampled the medical records by entering every 3rd file, n=1498 (33%), starting from a randomly chosen start position selected between 1-3. One duplicated file was removed leaving a total of 1,497 patient records included in this retrospective analysis (see study flow chart in S1 Fig).” See line 163-165.

The files were not sampled based on completeness of the records, and thus we believe that the missingness of this sub-sample is representative of the missingness across the records. 

11. Table 1; Insert exact number of available anthropometry rather indicating missing

[MUAC, WHZ, WAZ, HAZ] 

Response: This has been added thank you.

12. What the access and watch group of antibiotics with examples.

Response: The recent WHO classification has been better explained and examples have been added in both the methods (see response to question 8), to the footnote of Table 1 and to Table S7.

11. “Missingness in anthropometry was undocumented in approximately 40% of health records” this is the greatest limitation of this study and is missing the study limitation, kindly provide exact number that are using.

Response: We agree with the reviewer that this is a main limitation of the study especially considering that Moyo is a nutritional ward specialised in severe malnutrition. The lack of anthropometric recording demonstrates that procedures to document anthropometry should be improved. We aimed to discuss this as one of our main messages in the limitation section. To further clarify we added the following text in italics to the following section:

“The main limitation of the study is the high missingness rates in important anthropometric measurements. While we made diligent efforts to assess and account for missing data, the potential for reporting biases exist, e.g., healthcare providers possibly avoiding performing anthropometric measurements in sicker children, or the decrease in reporting of vaccination status. The review of archived records can also be useful in identifying lacks in clinical documentation that can be addressed, improving data capture would be valuable for both research and clinical practice.”

We have now also tallied the “available” data for each anthropometric score indicated by group as suggested by the reviewer. Please see Table 1. 

12. “This decline coincided with an increase in unreported vaccination status of patients” I can not locate this in the results?”

Response: The results were presented in Figure 2 and in Supplemental Table 6 (full model results). We have clarified the figure legend to mention that “unreported vaccination” are indicated by the red line within panel B (See figure legend of Figure 2). But, also, to better point the reader to the results, we have now added the panel “letter” within the text to indicate specific figure panels. This has now been done for all figures.

13. (i.e. 346 mortality cases out of 1498 children)-This appear incorrect since close to 40% of children did not have anthropometry for proper classification/inclusion/exclusion

Response: The classification of a patient as died during hospitalization does not depend on having a complete record for anthropometry. Thus, children with missing anthropometry were not excluded from these trend analyses and the mortality cases were clearly documented and, thus, we do not think there is error in this section due to recording biases regarding mortality. 

14. Table 2-can you clarify the models, model 2 did not include anthropometry???- unadjusted model 1 and model 3 for age, sex and HIV and model 2 further adjusted for Anthropometry 

Response: Thank you for this comment. We realised that the model description for the first column was missing. The first column presents the percentage and 95% confidence intervals of mortality for each group as derived from unadjusted logistic regression models. While the 2 other model columns present the Odds ratio of models adjusted by age, sex, and HIV (level-I adjustment) and by age, sex, HIV and WHZ (level-II adjustment). This has been clarified in Table 2 footnote and in the statistical section of methods.

Table 2 footnote now reads as follows: “1Percentage and 95% confidence intervals (95% CI) of mortality estimated for each group classification derived from unadjusted logistic regression models. Odds ratio (OR) and 95%CI are presented for 2models adjusted for age, sex, and HIV status and 3models additionally adjusted for WHZ.”

Method section, now reads as follows: “Percentage and 95%CI of mortality or readmission were calculated for each patient group (e.g., WHZ ≥ -3.5 or < -3.5; Age ≥ 5 years or ≥ 2 & < 5 years or < 2 years, edema yes/no; dehydration yes/no, diarrhea yes/no, vomit yes/no, cough yes/no, difficult breathing yes/no, HIV yes/no) and were derived from unadjusted models using the emmeans R-package. Odds ratio (OR) between groups were calculated from models adjusted for age, sex, and HIV status (Level-I adjustment); and additional models were further adjusted for WHZ (Level-II adjustment).” See line 144-149.

Discussion

15. “However, there was a significant 40% reduction in the prevalence of nutritional edema” insert n before the percent ?

Response: Due to this comment, we clarified the exact numbers in the result section, but kept the text as is in the discussion. The text in the result section now reads as follows: “Overall, 60% [95% CI, 57-62] (i.e., 854/1,497) of children were admitted with nutritional edema but prevalence decreased by approximately 40% over the 10 years (79% [95% CI, 72-85], 19/26 children, in 2011 to 40% [95% CI, 30-51], 20/49 children in 2021) (Fig 1C and S2 Table).” See line 170-172.

16. “To the best of our knowledge, this is the first study to cover a 10-year period and highlight changing trends in clinical presentation” This statement should be re-casted, this data has close to half missing with no details of what is missing in the anthropometry

Response: As mentioned above (question #13), children with missing anthropometry were not excluded from the trend analyses that were not specifically looking at anthropometry. Thus, the missingness of anthropometric data does not influence the classifications over the 10-year period of death or readmission, and does not influence the classification of most clinical features assessed including age, diarrhea, vomiting, edema, dehydration, cough, respiratory distress pallor, neurological signs, HIV status, etc.

However, considering this comment, we nuanced the sentence which now reads as follows: “To the best of our knowledge, this is the first study to highlight changing clinical trends over a 10-year period of child admissions to a nutritional rehabilitation unit in low resource settings. While changing trends in anthropometry were not found (potentially due to missingness and lack of power), we found that indicators of dehydration, respiratory distress, pallor, and neurological signs were more prevalent in 2021 compared to a decade earlier.” See line 280- 283. 

17. “We were unable to capture laboratory or radiological findings or include specific chronic conditions associated with malnutrition like tuberculosis due to low recorded prevalence” How low is this? This should have been reported, it is an important co-morbidity that may influence outcomes

Response: We were unable to systematically capture these findings as records were poor for the laboratory findings, and tuberculosis is not often tested for with less than 1% of cases having notes pertaining to suspected tuberculosis. This likely does not reflect the true prevalence.

Considering this comment, we edited the sentence as follows: “Laboratory findings were often poorly archived (e.g., thermal paper print outs with faded ink) and radiological assessments were not systematically done or archived (e.g., for tuberculosis diagnosis). We were unable to capture laboratory or radiological findings or include more rare chronic conditions associated with malnutrition like tuberculosis.” See line 330-332.

18. Figures 1 to 4 are poor qualities and hard to read, kindly revise to enhance readability

Response: Apologies if Figures 1-4 seem of poor quality, the exported dpi was at least 350 which is typically sufficient for line graphs. If the quality seems low, we suspect that the PDF export may not be incorporating the full resolution images. Thank you for flagging this issue, we will make sure it is resolved with the journal. 

19. S1 Table: The age and sex has non-linear trend, what inform the choice to report non-linear vs linear trend

Response: Thank you for this question, the explanation was indeed missing from the table footnote and methods section. Both linear and non-linear trends were tested and the decision to report possible/weak or stronger non-linear patterns was based on the effective degrees of freedom (EDF). EDF is estimated from generalized additive models and used to evaluate the significance of non-linearity where an EDF of approximately 1 suggests a linear relationship; an EDF greater than 1 but les

---

## [Editor Report · Decision Letter 1]

20 Sep 2024

Ten-year trends in clinical characteristics and outcome of children hospitalized with severe wasting or nutritional edema in Malawi (2011-2021): Declining admissions but worsened clinical profiles

PONE-D-24-21799R1

Dear Dr. Celine Bourdon,

We’re pleased to inform you that your manuscript has been judged scientifically suitable for publication and will be formally accepted for publication once it meets all outstanding technical requirements.

Kind regards,

Nigusie Selomon Tibebu, MSc

Academic Editor

PLOS ONE
---

## [Editor Report · Acceptance letter]

15 Oct 2024

PONE-D-24-21799R1 

PLOS ONE

Dear Dr. Bourdon, 

I'm pleased to inform you that your manuscript has been deemed suitable for publication in PLOS ONE. Congratulations! Your manuscript is now being handed over to our production team.

Kind regards, 

on behalf of

Assistant Professor Nigusie Selomon Tibebu 

Academic Editor

PLOS ONE